# Volumetric Absorptive Microsampling of Saliva for Pharmacokinetic Evaluation of Mycophenolic Acid and Its Glucuronide Metabolite in Pediatric Renal Transplant Recipients: Bioanalytical Method Validation and Clinical Feasibility Evaluation

**DOI:** 10.3390/ph18111744

**Published:** 2025-11-17

**Authors:** Arkadiusz Kocur, Joanna Sobiak, Agnieszka Czajkowska, Jacek Rubik, Tomasz Pawiński

**Affiliations:** 1Department of Drug Chemistry, Pharmaceutical and Biomedical Analysis, Medical University of Warsaw, Banacha 1, 02-097 Warsaw, Poland; tomasz.pawinski@wum.edu.pl; 2Department of Physical Pharmacy and Pharmacokinetics, Poznan University of Medical Sciences, Rokietnicka 3, 60-806 Poznan, Poland; jsobiak@ump.edu.pl; 3Therapeutic Drug Monitoring, Clinical Pharmacokinetics and Toxicology Laboratory, Department of Clinical Biochemistry, The Children’s Memorial Health Institute, Dzieci Polskich 20, 04-730 Warsaw, Poland; a.czajkowska@ipczd.pl; 4Department of Nephrology, Kidney Transplantation and Arterial Hypertension, The Children’s Memorial Health Institute, Dzieci Polskich 20, 04-730 Warsaw, Poland; j.rubik@ipczd.pl

**Keywords:** mycophenolic acid, mycophenolic acid glucuronide, saliva, renal transplantation, volumetric absorptive microsampling (VAMS)

## Abstract

**Background:** Mycophenolic acid (MPA) is frequently used in pediatric renal transplantation as part of immunosuppressive therapy, yet therapeutic drug monitoring (TDM) remains challenging. Accurate monitoring is essential due to MPA’s narrow therapeutic window, variable pharmacokinetics, and high protein binding. This study examined whether saliva could serve as a non-invasive alternative to plasma for measuring MPA exposure. **Methods and Results:** Concentrations of MPA and its primary glucuronide metabolite (MPAG) were determined in plasma, capillary blood, plasma ultrafiltrate, wet saliva, and dried saliva collected using volumetric absorptive microsampling (VAMS). A novel LC–MS/MS method for quantifying MPA and MPAG in dried saliva collected with the Mitra™ device was developed and validated within a 1–700 μg/L calibration range, demonstrating robust analytical performance. Dried and wet saliva showed high correlation (r = 0.99 and 0.98 for MPA and MPAG, respectively). However, both salivary matrices—dried saliva collected with Mitra™ (vsMPA, vsMPAG) and wet saliva (sMPA, sMPAG)—exhibited poor correlation with unbound (fMPA, fMPAG) and total plasma concentrations (tMPA, tMPAG). A modest, yet positive, correlation was observed between the measured concentrations for the following pairs: sMPA versus fMPA (r = 0.376, *p* = 0.1036), sMPA versus tMPA (r = 0.305, *p* = 0.1904), sMPAG versus fMPAG (r = 0.205, *p* = 0.3851), and sMPAG versus tMPAG (r = 0.472, *p* = 0.0012). Pharmacokinetic parameters supported these findings, highlighting discrepancies between saliva and plasma. **Conclusions:** From a clinical perspective, saliva sampling—although minimally invasive and patient-friendly—does not offer a reliable substitute for plasma in routine TDM of MPA and MPAG. Capillary blood collected through VAMS remains a promising alternative for long-term monitoring of pediatric patients; however, several considerations still need to be addressed.

## 1. Introduction

The use of saliva as an alternative biological matrix in biomedical research is increasingly important, especially in the fields of omics, toxicology, and therapeutic drug monitoring (TDM). Saliva is one of the most accessible biological matrices, and its collection is a minimally invasive procedure. Saliva has a slightly acidic pH (6.80–7.20) and is produced at a rate of 500–1500 mL daily, mainly by the major salivary glands. Its primary component is water (99%), with inorganic and organic compounds (proteins and non-proteins) making up the rest. Several types of saliva exist, classified by biochemical composition (serous, mucous, or mixed) and gland origin [1,2].

Additionally, during sampling, both stimulated and unstimulated saliva may be collected. Unstimulated saliva is collected without any action that increases salivary flow. It typically involves allowing saliva to naturally pool in the mouth and then collecting it through passive drooling or spitting into a sterile container. In patients who cannot spit, a plastic pipette may be used for administration. Stimulated saliva is collected after intentionally increasing saliva production. This can be done by chewing paraffin gum, a rubber band, a paraffin block, or a piece of parafilm for about one minute before spitting into a tube. Chemical stimulants, such as citric acid or sodium bicarbonate, may also be applied—for example, by using citric acid-soaked cotton swabs on the buccal mucosa and then collecting the saliva with a pipette. Even chewing a swab increases salivary flow, so it still counts as stimulated saliva [1,2,3,4,5,6].

Nonetheless, the main challenge with saliva is standardizing the sample collection process. The presence of numerous saliva collection devices on the market does not simplify this, as sample quality can vary depending on the device used [1,2,3]. Typically, saliva collection devices consist of cotton or synthetic swabs, paper strips, passive drool funnels or tubes (especially for unstimulated saliva), or chewing gums and materials enriched with citric acid (used for stimulated saliva) [3]. Recently, volumetric absorptive microsampling (VAMS) and quantitative dried blood spots (qDBS) have been recognized as alternatives to traditional venipuncture for blood collection. The Mitra™ microsampling device (Neotheryx/Trajan, Torrance, CA, USA) is primarily evaluated for analyzing multiple analytes in capillary blood. However, according to the manufacturer, this device can also collect various matrices, including saliva [4]. The Mitra™ device features a hydrophilic polymer tip that absorbs a fixed 20 µL volume of fluid with a quick touch to the sample surface. Each device consists of a plastic handle ending with a small polymeric tip, which is capable of collecting a fixed amount of matrix—10, 20 or 30 µL, with an RSD of less than 5%. The primary advantage of VAMS is its volumetric sample collection, which is essential for standardizing sampling procedures and validating analytical methods. Additionally, the dried matrix within the microsampling device ensures longer stability of analytes compared to wet samples [4,5].

Undoubtedly, the benefits of using saliva as an analytical matrix include non-invasive sampling, simple collection in patients with limitations and in children, low cost, the ability for samples to be collected by non-qualified personnel, donors themselves can collect samples, there is no health risk to the donor, and controlled sampling is possible, reducing the risk of adulteration. However, several disadvantages may also occur small sample volume, potential interference, low concentrations of analytes of interest in saliva, which can impede detection, the necessity for recent exposure to analyze certain substances, and the risk of an increased pH level in the matrix after stimulating saliva flow, which could lead to lower concentrations of analytes of interest [1,2,3,4,5,6,7].

For TDM, saliva is highly advantageous because it can reflect the free (unbound to plasma proteins) drug fraction and ideally correlate with the total drug concentration in the blood [8]. Since the free fraction causes ‘real’ pharmacological effects, using saliva as an alternative matrix appears beneficial in theory. Population-level relationships from plasma–saliva bridging studies can inform routine care when only salivary measurements are available. In such cases, total plasma drug concentrations may be estimated from salivary concentrations using a population-specific conversion formula [1,2,3,4,5,6,7]. This approach has been investigated in the literature, particularly with many antiepileptic drugs, analgesics, and anti-infectives, where the proportion of free drug relative to its total level is relatively high [6,8]. Comprehensive review studies have discussed the utility of saliva in TDM, and clinical biochemistry methods are documented in the literature [6,8]. Notably, several analytes, including cortisol and drugs of abuse, are well established for measurement in saliva. The success of cortisol monitoring in saliva is partly due to the standardization of the collection process.

Conversely, several factors can influence the salivary concentration of drugs, including saliva pH, the plasma protein binding rate, and the compound’s physicochemical properties (e.g., lipophilicity or polar surface area) [8,9]. One of the main challenges associated with using saliva as a biological matrix is the often-poor correlation between drug concentrations in saliva and those in plasma or blood. This limitation significantly reduces its potential usefulness for therapeutic drug monitoring (TDM). The diffusion of drugs from plasma into saliva is affected by several physiological factors, particularly intra-individual variations in salivary pH, which can significantly influence the partitioning and diffusion of weak electrolytes.

Mycophenolic acid (MPA) remains a vital immunosuppressant for pediatric kidney transplant patients because of its strong ability to inhibit lymphocyte proliferation. Nevertheless, MPA has a narrow therapeutic window, significant variability in pharmacokinetics both between and within individuals, and variable protein binding, all of which make dose optimization challenging. Insufficient drug levels increase the risk of graft rejection, while excessively high levels can lead to toxicity, such as gastrointestinal issues and hematological side effects. Therefore, therapeutic drug monitoring (TDM) is routinely recommended to achieve the right balance of efficacy and safety. Due to the difficulties of frequent blood sampling in children, there is growing interest in less invasive methods like saliva-based TDM. However, the high plasma protein binding and pH-dependent diffusion of MPA complicate the establishment of a reliable correlation between saliva and plasma. MPA is classified as a compound with high affinity for plasma proteins (~97–99%) and potential excretion into saliva. Some studies have shown a relatively strong correlation between MPA plasma levels and saliva concentrations for total MPA (tMPA) and free MPA (fMPA) concentrations. In contrast, others have reported a poor correlation [10,11,12,13,14,15,16,17,18]. Because the AUC (area under the concentration-time curve) is recommended as an appropriate pharmacokinetic (PK) parameter for routine TDM of MPA, applying non-invasive saliva sampling (with the possibility of self-sampling) becomes an attractive approach, especially in pediatric and adolescent populations [10].

Detecting drugs in saliva requires sensitive and selective analytical methods. For salivary MPA (sMPA), techniques such as HPLC-FLD (high-performance liquid chromatography with fluorescence detection) and LC-MS/MS (liquid chromatography-tandem mass spectrometry) have been reported, with a lower limit of quantification (LLOQ) of less than 3 µg/L [11,12,13,14,15,16,17,18,19]. Although saliva appears to be a relatively simple biological matrix due to its high water content, analyzing it can be challenging because of the presence of physiological microbiota, concurrently administered drugs and compounds, food particles, and occasionally blood. During method validation, the extraction of the analyte from the saliva collection device should be carefully assessed based on its recovery. Various factors can influence the pre-analytical, analytical, and post-analytical stages, including errors in sampling, extraction of the target compound from saliva, matrix effects, and the implications for assay results [19].

Saliva has been suggested as a minimally invasive method for monitoring MPA levels; however, its clinical use has been slow due to inconsistent results in various studies. Research on salivary MPA (sMPA) remains limited and varied, particularly among pediatric transplant patients. Several studies involving adult and pediatric kidney transplant recipients, as well as healthy individuals, have demonstrated strong correlations between saliva and plasma levels of total and unbound MPA (r ≈ 0.8–0.9), indicating a potential for therapeutic drug monitoring [12,13,14,15]. In some adult groups, average salivary MPA levels closely matched plasma concentrations, except during morning troughs [13,15]. However, other studies—particularly in patients taking enteric-coated mycophenolate sodium—found weak or no correlation between saliva and plasma levels [12,13,14,15], raising concerns about the consistency and clinical reliability of this method. These conflicting findings underscore the need for comprehensive research on saliva-based MPA measurement, particularly in children, and emphasize the importance of rigorous methodology and comparison with a standardized method for saliva collection.

The study aimed to develop and validate a novel LC–MS/MS method for quantifying MPA and MPAG (mycophenolic acid glucuronide) in dried saliva samples collected using the VAMS Mitra™ device. Additionally, the pharmacokinetics of MPA and MPAG were assessed and compared across multiple matrices, including liquid (sMPA and sMPAG) and dried saliva (vsMPA and vsMPAG), plasma (‘total’; tMPA and tMPAG), capillary blood (vMPA and vMPAG), and plasma ultrafiltrate only (‘free’; fMPA and fMPAG). To the best of our knowledge, this is the first study to implement a comprehensive approach for TDM of mycophenolate across different biological matrices, while introducing a novel VAMS strategy for saliva collection in evaluating MPA pharmacokinetics in pediatric renal transplant recipients.

## 2. Results

### 2.1. Results of Saliva Sampling Correctness (Mitra™ VAMS Device) and Analytes Extraction Evaluation

The mean mass of “wet” saliva obtained from six independent sources was 0.9913 ± 0.035 g/mL (CV = 3.12%), while that of the ten evaluated VAMS samples was 1.0012 ± 0.021 g/mL (CV = 2.20%). These results align with previously reported saliva density values (~1.003 g/mL). No deviations were observed regarding the contact time of the VAMS tip with the matrix, in agreement with the findings of Mercollini et al. [5].

The drying time for the collected samples was evaluated at three intervals: 1 h, 2 h, and 4 h at room temperature (50% relative humidity). No deviations in vsMPA or vsMPAG recovery values were observed at the tested LQC level after IS correction (103.15–98.99% for MPA and 104.15–100.09% for MPAG, n = 3). Based on these results, 1 h was selected as the most efficient and practical drying time for sample processing. 

Various extraction solvents (methanol, acetonitrile, and water in different ratios) were tested to optimize analyte recovery from the Mitra™ device. Additional process improvements included vortexing (15 min), sonication (15 min at 36 °C), and freeze incubation (−40 °C for 15 min). The best extraction was achieved using a water/acetonitrile/methanol mixture (5:4:1, *v*/*v*) combined with all three supporting steps. Under these conditions, recoveries significantly increased—from 89.99% to 101.00% for MPA and from 84.41% to 98.22% for MPAG at the LQC level.

### 2.2. Analytical VAMS-Based Method Development and Optimization

The chromatographic separation of MPA and its metabolite (MPAG) is crucial for ensuring reliable results and quality. The metabolite is unstable in the mass spectrometer ion source and tends to dissociate into MPA. Consequently, various chromatographic columns with gradient flow of mobile phases were tested. The best peak shapes, shortest runtime, reproducibility, and effective separation of compounds were achieved using the Phenyl-Hexyl column in gradient mode, as described in Section 2.4. Buffering of the mobile phase with acetic acid and ammonium acetate was optimized experimentally by analyzing peak shape, area, and matrix interferences. Representative chromatograms for blank samples and LLOQ are included in the Appendix A. The retention times for the compounds were 3.10 min and 3.40 min for vsMPAG and vsMPA (coeluted with d_3_-MPA), respectively. The optimal mass spectrometry conditions were established using a tuning mixture of MPA, MPAG, and MPA-d_3_. Infusion experiments with 0.1% formic acid in a 1:1 (*v*/*v*) mixture of water and methanol, along with 2 mM ammonium acetate, facilitated the adjustment of compound-dependent parameters (EP, CXP, CE, DP) and ion source settings, resulting in stable signals and reproducible ionization. MRM transitions and other mass spectrometry parameters are provided in Section 2.4.

### 2.3. Validation Results of Dried Saliva—VAMS-Based Method

The LC-MS/MS method for quantifying vsMPA and vsMPAG in saliva samples collected using the Mitra™-VAMS device was validated within a calibration range of 1–700 μg/L, in accordance with international guidelines [20,21]. Calibration curves were generated by plotting the analyte-to-IS peak area ratio against the known concentrations of the analyte. The process involved applying linear regression with 1/x and 1/x^2^ weighting, and the average R^2^ values were 0.9958 (CV% = 0.63%) and 0.9930 (CV% = 0.89%) for vsMPAG, respectively. The average calibration equations are y = 0.1098x + 0.0098 for MPA, and y = 0.1453x − 0.0882 for MPAG, where y represents the analyte-to-IS peak area ratio, and x denotes the nominal analyte concentration. The LOD value (0.25 μg/L) was determined experimentally by injecting the diluted LLOQ standard and observing the signal-to-noise ratio (S/N > 50, used as a cut-off level). The overall mean precision for the validated method was determined to be 5.804% for MPA and 5.347% for MPAG. The summarized back-calculated values of intra- and inter-run accuracy and precision for both analytes at LLOQ, LQC, medium quality control 1 (MQC1), medium quality control 2 (MQC2), and HQC levels are shown in Table 1. All calculated values meet the specified acceptance criteria (Section 2.7). In almost all analyzed cases, the IATDMCT recommendation regarding immunosuppressant quantification (precision ≤ 10%) has also been met [22].

The carry-over met the EMA standards for vsMPA, vsMPAG, and IS, with values of 0.984 ± 0.416%, 1.003 ± 0.518%, and 0.223 ± 0.082%, respectively [22]. Additionally, no interference from concurrently administered drugs, unknown compounds, or matrix was observed throughout the entire chromatographic run (peak area of interferences lower than 20% for the analyte (LLOQ) and less than 5% for the IS responses).

No ion enhancement or suppression was observed, as IS correction effectively compensated for potential interferences. The calculated ME, AR, and PE values are shown in Appendix A. Variability in matrix interferences was observed across different dried saliva sources, reflecting donor-dependent differences in saliva composition.

Dried saliva samples remained stable for at least two weeks under all tested storage conditions, including room temperature (20–22 °C; 40–60% of relative humidity) with or without light exposure. In all cases, stability values were within the EMA acceptance criterion (±15% of the initial concentration) [22]. The detailed results are given in Appendix A. No significant (>15%) deviation in the mean initial value was observed after 24 and 72 h following the initial analysis for samples stored at 15 °C in the autosampler.

### 2.4. Clinical Outcome

The validated method for vsMPA and vsMPAG determination in dried saliva samples was applied clinically to 20 pediatric patients undergoing renal transplantation. Concurrently, tMPA and tMPAG in plasma, vMPA and vMPAG in capillary blood, and sMPA and sMPAG were measured in samples collected at 0, 0.5, 1, 2, and 4 h post-dose for each patient. Additionally, in plasma ultrafiltrate obtained by ultracentrifugation, fMPA and fMPAG were quantified. Individual PK profiles for all patients are provided in the Appendix A. Selected clinical and demographic parameters for the study group are listed in Table 2.

Results of MPA and MPAG concentration determinations in each type of collected matrix are presented as Appendix A. A limited but significant relationship was observed between sMPA and fMPA (r = 0.376, *p* = 0.1036), sMPA and tMPA (r = 0.305, *p* = 0.1904), sMPAG and fMPAG (r = 0.205, *p* = 0.3851), and sMPAG and tMPAG (r = 0.472, *p* = 0.0012), as estimated using Spearman’s correlation coefficient, given the non-normal distribution of the data. In contrast, a relatively strong correlation was observed between paired results: fMPA versus tMPA, vMPA versus tMPA, fMPAG versus tMPAG, and vMPAG versus tMPAG (r = 0.883, *p* < 0.0001; r = 0.979, *p* < 0.0001; r = 0.717, *p* = 0.0004 and 0.891, *p* < 0.0001, respectively). Furthermore, a high degree of correlation was also noted for the sMPA versus vsMPA and sMPAG versus vsMPAG comparisons (r = 0.952, *p* < 0.0001; r = 0.974, *p* < 0.0001, respectively). A complete visualization of the correlation between MPA and MPAG levels in the tested matrices is provided in the Appendix A. 

### 2.5. Cross-Validation

The equivalence between “wet” and dried salivary concentrations of MPA and MPAG was assessed through Passing–Bablok regression, Bland–Altman bias estimation, and correlation analysis. The matrices were deemed interchangeable because the acceptance criteria were met: the confidence intervals for the intercept (A) and slope (B) included 0 and 1, respectively. The calculated percentage bias was 3.30% for MPA and 6.49% for MPAG. For MPA, both the LoA (bias < ±20% in at least 67% of paired samples) and the CoA (bias < ±15% in at least 67% of paired samples) were achieved. For MPAG, the LoA criterion was met, while the CoA was narrowly missed (66% of paired samples < ±15% bias; acceptance criterion: 67%). Detailed results are shown in Table 3, with graphical regression and bias analyses displayed in Figure 1. The compared values demonstrated a strong relationship, with a correlation coefficient (for both analytes) exceeding 0.98.

### 2.6. Pharmacokinetics Application

The non-compartmental PK analysis has been performed for both MPA and MPAG analytes in 20 pediatric renal transplant recipients for each included matrix. The AUC_0–4_, T_max_, and C_max_ parameters were calculated using the R (version 4.5.1) package software (R Core Team, 2025), supported by RStudio (version 2025.05.1+513) from Posit PBC software (Boston, MA, USA). Additionally, the unbound fractions of MPA and MPAG have been calculated as fMPA-to-tMPA and fMPAG-to-tMPAG percentage ratios, respectively. The vMPA and vMPAG capillary blood levels were corrected using formulas validated in our department (partially published by Kocur et al.) and expressed as estimated capillary plasma concentrations [21]. Mean values of the estimated parameters, along with their corresponding correlation coefficients, are presented in Table 4. Summarized profiles of MPA and MPAG for all patients, grouped by the biological matrix used, are presented in Appendix A, respectively. Individual PK profiles for each patient included in the study are provided in the Appendix A.

During PK analyses, the plasma protein binding of MPA and its metabolite MPAG was determined. The mean binding was 98.83% for MPA and 81.15% for MPAG.

The correlation between individual AUC_0–4_ values for each tested matrix type, normalized to mycophenolate dose, was more significant than the connection between matrix type-dependent concentration values (presented in Appendix B and Appendix A). The calculated Spearman correlation coefficients (r) for AUC_0–4_ normalized to the dose of MMF for MPA and MPAG are provided in Appendix B. After normalization, the correlation coefficients demonstrated a more significant (but still limited) correlation of normalized AUC_0–4_ values (for MPA and MPAG) between saliva and in other matrices, such as plasma and capillary blood (VAMS samples). Additionally, a positive, moderate correlation has been demonstrated between the free fraction of MPA and its metabolite, as well as corresponding salivary levels.

### 2.7. Clinical Feasibility Evaluation

During the clinical phase, patients completed self-efficacy questionnaires about the sample collection process. Seventy per cent preferred the home-based self-sampling method, and 75% of participants found saliva collection easy to perform. Furthermore, 90% expressed a preference for the VAMS technique. To date, 40 patients have been enrolled in the presented PK study; however, 20 were excluded from PK analysis due to difficulties, such as a gag reflex during sampling, which made saliva collection impossible with the Salivette^®^ device, or refusal to provide saliva with any device.

## 3. Discussion

In the presented study, the novel method for determining vsMPA and vsMPAG in dried saliva VAMS samples has been validated using LC-MS/MS analysis according to guidelines [23,24,25,26]. The application of the Mitra™ technique for saliva collection has been demonstrated in the literature for perampanel [27], antidepressants [28], and cannabinoids and their metabolites [29]. In contrast, our method is the first to use VAMS for saliva collection for MPA and MPAG. The estimated analyte concentration in saliva determined the necessary assay. The LC-MS/MS is the most appropriate method due to its sensitivity, but simpler techniques have also been employed in wet saliva, such as HPLC-FLD for MPA [10,17,18].

Saliva collection using traditional devices is non-invasive; however, the collected volume is only approximate, requires centrifugation, and the wet matrix cannot be shipped by post in decentralized care settings. Our study confirmed the feasibility of saliva collection. It demonstrated that the dried matrix can be reliably processed by post due to the satisfactory stability of the tested analytes in the Mitra™ device.

Many factors affect the quality and composition of collected saliva samples, which justifies the need for standardizing the sampling process. First, the collection of stimulated or unstimulated saliva can be performed. For some analytes, a stronger linear relationship was demonstrated between plasma and salivary levels in stimulated saliva samples [6,8]. In the presented study, we have decided to collect unstimulated saliva because it is easier to obtain in pediatric patients. The idea of collecting stimulated saliva was abandoned due to discomfort reported by some patients, such as a gag reflex or an unpleasant taste of citric acid.

Another factor to consider is the saliva composition, as the amount of water and salivary mucins can affect the distribution of analytes and their extraction from the saliva collection device. Some inconsistencies may be observed between spiked saliva samples (CCs and QCs) and objective clinical samples. Therefore, we decided to incubate spiked saliva samples at 36 °C for 30 min before absorption into the Mitra™ VAMS collection device to simulate the in vivo conditions of the sampling process. Following previously published suggestions, the sonication or freeze-thawing of drug-free saliva samples appears to be beneficial for obtaining a homogenous sample [10,11,12,13,14,15,16,17,18,19]. In the case of the Mitra™ application, the interferences caused by saliva viscosity and mucins are omitted because the sample is in a dried form. It is also essential to properly evaluate the impact of potential differences related to matrix effects using samples from an appropriate number of individual saliva sources.

In the literature, various extraction rates of analytes (MPA and MPAG) from Salivette^®^ swabs have been reported. Wiesen et al. demonstrated a recovery greater than 95% from synthetic swabs, while Paniagua-Gonzalez et al. reported other findings for cotton swabs (60–65% of MPA recovery) [30,31]. In contrast, Sobiak et al. evaluated recoveries for MPA and MPAG within an acceptable range of 94–105% [17].

The population of pediatric patients included in the present study required a combination of various drugs, including antimicrobials that affect human microbiota and those that influence overall hydration levels. Therefore, variability in oral microbiome composition is another factor contributing to differences in the composition of saliva samples. It is crucial to consider metabolites (i.e., MPAG), which can be converted back to MPA through bacterial deconjugation. Additional factors, such as hydration status and oral cavity condition, influence the pH of saliva and, consequently, the distribution of the analyte of interest in saliva. These factors should be carefully analyzed to establish the suitability of saliva as a surrogate matrix in routine TDM of MPA.

A recently published review study comprehensively described the relationships between the physicochemical properties of drugs and their distribution in saliva [6]. In the case of acidic medications (such as MPA), the authors demonstrated a statistically significant relationship between a lower ionization rate and a higher affinity to plasma proteins. In the case of MPA, authors of the review stated that TDM based on saliva is possible; however, only the Ferreira et al. study has been cited, as it fulfilled the inclusion criteria for a statistical evaluation [6,16]. A relatively high correlation has been demonstrated between salivary and plasma MPA levels in the mentioned study [16].

On the other hand, there are some controversies regarding potent relationships, which were also highlighted in the IATDMCT Consensus [10]. For example, studies by Brooks et al. and Cossart et al. demonstrated limited correlation between MPA and MPAG levels in plasma, saliva, and plasma ultrafiltrate [11,14]. However, in the presented work, a relatively low correlation has been observed between MPA and MPAG levels in saliva (versus other matrices). The correlation coefficient between AUC_0–4_ values has been improved after normalization to the individual dose of MMF.

The results of PK parameters for children enrolled in the presented study are similar to those reported for adult renal transplant recipients in the study by Cossart et al. [11]. A limited correlation has been observed between salivary and fMPA, as well as between salivary and fMPAG, and between salivary and plasma levels of analytes, as confirmed by our research. Brooks et al. reported similar findings, but in the population under EC-MPS (enteric-coated mycophenolate sodium) formulation [14]. However, the relationship between tMPA and fMPA concentrations was relatively strong (r = 0.890), whereas poor correlations were observed for sMPA levels compared with tMPA and fMPA levels (r = 0.510 and 0.410, respectively) [14].

Some studies reported a strong correlation between tMPA, fMPA, and sMPA concentrations. In the study by Mendoza et al., which included 11 adult renal transplant recipients, the correlation coefficients were 0.909 and 0.910 for sMPA levels compared with tMPA and fMPA levels, respectively [13]. Ferreira et al. also reported a strong correlation, not only between MPA concentration levels (saliva vs plasma), but also for AUC_0–12_ in both tested matrices for MPA and MPAG [16]. A potential strong relationship was also reported by a previous study by Sobiak et al. regarding the pediatric population with nephrotic syndrome treated with MMF [17,18].

It should be highlighted that the salivary level of the drug should reflect the unbound (free) drug level in plasma. Typically, the unbound fraction of MPA ranges from 1 to 2.5%. In the case of MPAG, the binding to albumin ratio is approximately equal approximately 82% in stable transplant recipients; thus, in the evaluated population, the mean bound fraction was 81.15%. In patients with severe renal impairment, the concentration of the major MPA metabolite, MPAG, may increase up to 3–6-fold. This increase in MPAG leads to the displacement of MPA from its binding sites, resulting in a 7% increase in the MPA-free fraction [9,10]. None of the patients represented liver dysfunction; therefore, significant deviation of the target value was not observed.

Peak plasma concentrations of MPA and MPAG typically occur 1–2 h and 1–4 h after administration, respectively [9,10]. According to the literature, the values of the PK parameters of MPA and MPAG tend to stabilize during the first year after transplantation. In the presented study, the values of that parameter ranged from 0.5 to 2 h (0.5–1.0 for tMPA) for both analytes in the tested matrices. In some pediatric population studies regarding PK, similar observations have been reported [9]. Typically, MPAG concentrations are 20–100 times higher than those of MPA [9], a finding also demonstrated in the presented study. For AUC_0–4_ (normalized to drug dose), the mean value of tMPAG was approximately five times higher than that of tMPA. In Ferreira et al. study, the median MPAG AUC_0–12_ was ten times higher than the median MPA AUC_0–12_ [16].

A notable study by Alsmadi et al. has been published, featuring complex population physiologically based pharmacokinetic (pop-PBPK) modeling of MPA individual exposure in renal transplant recipients, based on compartmental levels of the drug (plasma, saliva, and renal tissue) [12]. Future studies, considering the utility of saliva in TDM of mycophenolate, should incorporate similar models that predict salivary drug excretion.

The limited number of samples (5 samples for each matrix collected at 0–4 h time points) in the PK profile may be considered a limitation of the presented study. However, it is a proof-of-concept study, and the primary aim was to assess the application of the Mitra™ microsampling device in saliva collection within clinical settings, including pharmacokinetics studies. Studies were performed in the Renal Transplantation Outcome Clinic because it included stable pediatric renal recipients. Therefore, due to logistical reasons, the shortened protocol for sampling before PK analysis of the profile in the 0–4 h range was considered. The 0–4 h sampling window was chosen to capture the absorption and early distribution phases after morning dosing, which show the most dynamic concentration changes and are suitable for pediatric clinical visits. The goal was not to achieve complete drug elimination; instead, this timeframe enabled the practical collection of paired plasma, capillary VAMS, and saliva samples with minimal burden on young patients. Extending sampling beyond 4 h would require longer clinic stays and repeated saliva stimulation, which are difficult in young transplant recipients and raise ethical concerns. While this approach does not encompass the full AUC, it aligns with studies that focus on matrix agreement rather than comprehensive pharmacokinetic profiling. Similarly, in the Cossart et al. study, saliva (and conventional plasma) was collected at C_0_, C_1_, C_2_, and C_4_ time points [11]. Conversely, only three samples (C_0_, C_0.5_, C_2_) were collected in the Wiesen et al. study, but the AUC_0–12_ was estimated using Bayesian methods [30]. In other studies, more samples were taken within the 0–12 h range for complete AUC estimation [12,13,14,15,16,17,18]. From a pharmacokinetic perspective, the use of saliva for routine TDM of MPA is limited by the lack of target reference ranges for both the threshold concentration and AUC_0–t_ in the literature.

The interchangeability between two analytical approaches should be demonstrated using paired samples from at least 40 individuals, as recommended. In the present study, we opted to illustrate the correlation using all 100 samples from 20 patients. While this may be considered a limitation, the strong correlation observed supports the use of the microsampling device to standardize saliva sampling in extensive population studies. Initially, 40 patients were enrolled, but half withdrew due to issues related to the saliva collection process (discussed in Section 2.7).

The VAMS is suitable for collecting a small amount of saliva, especially in patients with xerostomia or other diseases affecting salivary excretion. However, the volume collected using Mitra™ in the presented study is low, and similar interferences to wet saliva collection may occur, such as blood contamination. In the evaluated samples, blood contamination in the saliva was only visually observed, which may be a limitation. To assess the potential impact of blood contamination on MPA, the salivary levels of transferrin (used as a biomarker for blood contamination) can be measured. For instance, Mendoza et al. used a Salimetrics transferrin kit to detect transferrin and excluded samples with a level greater than 1 mg/dL. Fasting wet saliva samples showed significantly higher transferrin levels compared to non-fasting samples, and this was associated with increased MPA concentration [13].

## 4. Materials and Methods

### 4.1. Standards, Chemicals, and Laboratory Supplies

MPA certified reference standard (1.0 mg/mL in acetonitrile) and certified reference internal standard (d_3_-MPA, 0.1 mg/mL in acetonitrile) were obtained from Merck (Darmstadt, Germany). The MPAG standard was supplied by MedChem Tronica (Sollentuna, Sweden). The zinc sulfate heptahydrate (ZnSO_4_·7H_2_O) was from Supelco (Bellefonte, PA, USA). LC-MS grade organic solvents (methanol, acetonitrile, and 2-propanol), LC-MS grade water, and mobile phase modifiers (formic acid and ammonium formate, both of which are LC-MS grade) were purchased from Merck (Darmstadt, Germany). Drug-free saliva, whole blood, plasma, and its ultrafiltrate were obtained from healthy adult volunteers.

The Salivette^®^ (cotton swab) device for saliva collection was supplied by Sarstedt (Nümbrecht, Germany), while the microsampling device—Mitra™ (20 µL) was acquired from Neoteryx (Torrance, CA, USA) via Bioanalytic (Gdansk, Poland). The Verex™ filter chromatographic vials (PTFE membrane) were purchased from Phenomenex (Torrance, CA, USA). Automatic Reference™ pipettes, Multipette Repeater™ M4, and other plastic laboratory materials (test tubes, pipette tips) were provided by Eppendorf (Hamburg, Germany).

### 4.2. Stock, Working, Calibration, and Quality Control Solutions

The certified reference standard of MPA (1 mg/mL) was serially diluted by 10-, 100-, and 1000-fold to prepare working solutions for calibrator (CC) generation. The other ampoule of MPA standard was similarly diluted and used for quality control (QC) production. Two separate MPAG solutions were prepared using different weights to generate 1 mg/mL methanolic solutions. In the dried saliva VAMS-based method, the nominal concentrations of CC and QC are shown in Table 5. Briefly, the CC and QC samples were prepared as follows: 40 μL of drug-free saliva, collected with the Salivette^®^ device, were spiked with 10 μL of the working solution to achieve the desired final concentration. Then, 20 μL of the sample was absorbed using the Mitra™ device (at a 45° angle, for 2 s) and left to dry for 1 h.

The internal standard (IS) solution was prepared using a reference 0.1 mg/mL d_3_-MPA solution to achieve a corresponding concentration of 20 μg/L (20 × LLOQ; calculated per 20 μL of dried saliva). The protocols for CC and QC preparation for fMPA, fMPAG, tMPA, tMPAG, sMPA, and sMPAG analysis were based on previously published method validation studies carried out in our laboratories (Łuszczyńska et al., Kocur et al., Sobiak et al., respectively) [17,20,21].

### 4.3. Sample Preparation Protocols

#### 4.3.1. Liquid Saliva (sMPA and sMPAG)

The concentrations of sMPA and sMPAG in saliva were determined using a previously described LC-MS/MS method [17]. Briefly, 100 µL of saliva was used for analysis. To each sample, 50 μL of acetonitrile (for study samples) or working solutions (for calibration standards) and 50 μL of IS (d_3_-MPA for MPA determination) were added. Samples were vortexed and evaporated to dryness. The dry residues were reconstituted in 100 μL of a mobile phase consisting of a 1:4 (*v*/*v*) mixture of methanol and water, both of which contained 0.1% formic acid. After double centrifugation, the samples were analyzed using a Shimadzu Ultra-Performance Liquid Chromatography (UPLC) system (Shimadzu Co., Kyoto, Japan) coupled with a triple quadrupole mass spectrometer (LCMS-8030; Simadzu Co., Kyoto, Japan). The method was linear over the concentration range of 2–500 μg/L for both sMPA and sMPAG. This part of the work was carried out with the support of the Centre of Innovative Pharmaceutical Technology, Poznan University of Medical Sciences (Poznan, Poland).

#### 4.3.2. Dried Saliva (vsMPA and vsMPAG)

The MPA and MPAG levels in dried saliva (vsMPA and vsMPAG) samples (20 µL) collected using the Mitra™ microsampling device were processed as follows: the VAMS tip was removed with tweezers and placed in a 2 mL Eppendorf tube containing 400 µL of water/acetonitrile/methanol mixture (5:4:1, *v*/*v*) with an IS (d_3_-MPA). Samples were sonicated (36 °C, 30 min), vortexed (15 min), and frozen (−20 °C, 15 min). After centrifugation (10 min, 25,155× *g*), 350 µL of supernatant was transferred to a total recovery borosilicate vial (Finneran, Vineland, NJ, USA) and evaporated to dryness under a stream of nitrogen (40 °C) using Ultravap^®^ Levante (Porvair, Hampshire, Great Britain). The residue was reconstituted in 150 µL of mobile phase A, filtered in a Verex™ vial, and 5 µL was injected into the LC–MS/MS system. Analyses were performed at the Department of Drug Chemistry, Pharmaceutical and Biomedical Analysis, Medical University of Warsaw, Poland.

#### 4.3.3. Plasma Unbound Fraction (fMPA and fMPAG)

The unbound concentrations of MPA (fMPA) and MPAG (fMPAG) in EDTA plasma have been quantified using a modified method published by Łuszczyńska et al. [20]. Briefly, the 50 μL of plasma ultrafiltrate obtained using centrifugation (20 min., 36 °C at 13,500× *g*) of EDTA-plasma in an Amicon™ Ultra device (Merck, Darmstadt, Germany; 50 kDa cut-off) was added to the mixture of 10 μL of MPA—d_3_ working solution (1000 μg/L) and 300 μL of acetonitrile in a 1.5 mL Eppendorf tube. Next, the sample was centrifuged at 3000× *g* for 5 min at ambient conditions, and the supernatant was transferred into a chromatographic vial. Sample processing was performed at the Department of Drug Chemistry, Pharmaceutical and Biomedical Analysis, Medical University of Warsaw, Poland.

#### 4.3.4. Total Plasma Levels (tMPA and tMPAG)

For the determination of the total concentration of MPA (tMPA) and MPAG (tMPAG), our previously published method was used [21]. Briefly, 50 μL of plasma was diluted with 90 μL of deionized water, and 10 μL of IS (d_3_-MPA) was added. Next, 400 μL of the precipitation mixture (a 0.1 M aqueous zinc sulfate solution and acetonitrile, 50:50 (*v*/*v*)) was added. The sample was then shaken for 10 min at room temperature using an automatic rotary vortex (Ohaus, Parsippany, NJ, USA) and centrifuged for 10 min at 13,500× *g* at 4 °C thereafter. The supernatant was transferred into chromatographic vials and analyzed using LC-MS/MS. Sample processing was performed at the Department of Drug Chemistry, Pharmaceutical and Biomedical Analysis, Medical University of Warsaw, Poland.

#### 4.3.5. Capillary Blood Level (vMPA and vMPAG)

For capillary blood (obtained from the small blood vessels (capillaries) close to the surface of the skin after skin pricking), MPA (vMPA) and MPAG (vMPAG) determination were performed using VAMS tips (10 μL) loaded with the corresponding CC or QC. Subsequently, the tip (after at least 1 h of drying) was placed in an 1.5 mL Eppendorf-type test tube (Eppendorf AG, Hamburg, Germany) containing 150 μL of pure deionized water. Next, the test tube was shaken at room temperature for 1 h using the automatic rotary vortex. Subsequently, 10 μL of IS solution and 150 μL of the precipitation mixture were added. The sample was then processed as described in Section 4.3.4. Sample processing was carried out at the Department of Drug Chemistry, Pharmaceutical and Biomedical Analysis, Medical University of Warsaw, Poland [21].

### 4.4. LC-MS/MS Conditions for Dried Saliva Samples Determination

For LC-MS/MS detection, the Exion LC AE system paired with a 6500 triple quadrupole mass spectrometer (both from AB Sciex, Framingham, MA, USA) was used.

For mass detection, multiple reaction monitoring (MRM) transitions with electrospray ionization (ESI) in positive mode were employed. MPA was quantified using the 321.10 → 159.00 m/z transition and identified with the 321.10 → 207.00 m/z transition. The MPAG was monitored through the transitions 514.1 → 159.1 m/z (quantitative) and 514.1 → 207.00 m/z (qualitative). For IS (MPA-d_3_), the transition 324.3 → 159.10 m/z was monitored. The declustering potential (DP), entrance potential (EP), collision energy (CE), and collision cell exit potential (CXP) were optimized experimentally for each transition. Specifically, DP was set to 100 V, EP to 5 V, and CXP to 20 V, while CE values varied depending on the quantitative and qualitative transitions. The ion source was operated with the following parameters: curtain gas at 40 psi, collision gas set to “high”, voltage at 5500 V, source temperature 300 °C, GS_1_ at 60 psi, and GS_2_ at 30 psi. All transitions were acquired with a dwell time of 50 milliseconds.

To ensure effective chromatographic separation, a Luna Phenyl-Hexyl (100 Å, 100 × 2.0 mm, 3 µm) column from Phenomenex (Torrance, CA, USA) was selected. An Ultra Guard C_18_ Cartridge (also from Phenomenex), along with a compatible holder, protected the column. During analysis, the column was maintained at 45 °C in the oven with a total flow rate of 0.4 mL/min in gradient mode. The mixed mobile phase was generated using two independent solutions, (A) water buffered with acetic acid and ammonium acetate (0.2% and 4 mM, respectively) and (B) a methanol–acetonitrile mixture (50:50, *v*/*v*). The total run time was set at 5 min with the following stages in gradient mode: 5% of (B) from 0.00 to 0.75 min, linear gradient from 60% of (B) to 95% of (B) to 3.15 min, maintenance of 95% (B) between 3.51 min and 4.15, next re-equilibration to initial conditions. The mixture of methanol, acetonitrile, water, and propan-2-ol (1:1:1:1, *v*/*v*/*v*/*v*) was used for needle purging in external-internal mode. The sample injection volume was set to 5 μL.

### 4.5. Patients and Clinical Sampling Protocol

The clinical component of the study was conducted at the Kidney Transplant Outpatient Clinic of the Children’s Memorial Health Institute in Warsaw, Poland. In this study, twenty stable pediatric renal transplant recipients treated with the MPA prodrug since the date of transplantation (pharmacokinetically at steady-state), mycophenolate mofetil (MMF; CellCept^®^, Roche, Basel, Switzerland), were enrolled between June 2023 and December 2024. The therapeutic immunosuppressive scheme also included tacrolimus and glucocorticosteroids. More demographic data regarding to study group is presented in Table 5 (Section 2.4).

From each patient (if they consented to all sampling types), four samples were collected simultaneously at all defined time points of the dynamic PK profile (0, 0.5, 1, 2, and 4 h after drug administration). Whole blood was collected using standard venipuncture by a phlebotomist into a K_3_-EDTA test tube (Sarstedt, Nümbrecht, Germany). Plasma for analytical determination was obtained by centrifuging a portion of the whole blood and stored at −40 °C until assay. At the same time, 10 µL capillary blood samples were collected from a finger prick using the Mitra™ device (VAMS samples were stored in light-protected aluminum zip bags with desiccant at room temperature).

Additionally, the unstimulated saliva samples were simultaneously collected using the Salivette^®^ device (Sarstedt, Nümbrecht, Germany) and Mitra™ 20 µL microsampling tips (Neoteryx/Trajan, Torrance, CA, USA (all saliva samples simultaneously to plasma and capillary blood sampling: 0, 0.5, 1, 2, and 4 h after drug administration). Patients had not eaten for at least 30 min before saliva collection, and their mouths were rinsed with water. The study was conducted in accordance with the Declaration of Helsinki and the International Council for Harmonization Guideline for Good Clinical Practice. The protocol was approved by the Bioethics Committee of the Children’s Memorial Health Institute, Warsaw, Poland (approval No. 15/KBE/2023, with subsequent amendments). Written informed consent was obtained from all participants and their legal guardians.

### 4.6. Method Validation

The development and validation of the LC–MS/MS analytical method for determining vsMPA and vsMPAG collected with the VAMS-Mitra™ system were carried out under European Medicines Agency (EMA) guidelines, based on the ICH M10 recommendations and documents [22]. Furthermore, the recommendations concerning the analytical determination of immunosuppressive drugs published by the International Association of Therapeutic Drug Monitoring and Clinical Toxicology (IATDMCT) were also implemented [23,24]. 

To evaluate the acceptability of the analytical method and the reliability of the results, the following parameters were examined: specificity, selectivity, linearity, limit of detection (LOD), LLOQ, accuracy, precision, carry-over effect, reinjection reproducibility, and stability. In assays, the stable-isotope-labeled internal standard (SIL-IS) of MPA (d_3_-MPA) was applied according to the guidelines’ recommendations [22,23,24,25].

For evaluating selectivity and specificity, both double blanks (drug-free dried saliva samples without SIL-IS) and zero calibration samples (blank; drug-free dried saliva samples with SIL-IS added) were analyzed to identify any interferences from concomitant drugs, their metabolites, or matrix compounds. The guidelines specify that selectivity is considered acceptable if the detector response to interfering components is less than 20% for the analyte (LLOQ) and less than 5% for the IS.

For calibration and linearity assessment, a linear 1/x weighting method was applied across the concentration range of 1–700 μg/L. Each calibration curve was created by plotting the MPA (or MPAG)/SIL-IS peak area ratio against the assigned concentration. Nominal concentrations of CCs, derived from the spiked matrix, are detailed in Section 2.2. For each matrix type, fifteen calibration curves were prepared and analyzed using a spiked matrix approach. Recommended thresholds for correlation coefficient and coefficient of determination were R ≥ 0.990 and R^2^ ≥ 0.995, respectively. QC samples were used to assess within-run and between-run accuracy, expressed as the mean ± standard deviation (SD), and precision, expressed as the coefficient of variation (CV%). Measured concentrations were expected to be within ±15% of the nominal value, or ±20% at the LLOQ.

The carry-over was assessed over 10 runs by injecting the high QC (HQC) sample (500 µg/L) immediately before the double-blank sample. The lack of this effect is identified when the analyte and IS responses in the blank sample are less than 20% and 5% of the responses at the LLOQ, respectively.

For matrix effect (ME) evaluation, the protocol proposed by Matuszewski et al. [26] has been used. Additionally, process efficiency (PE) and absolute recovery (AR) were assessed. Three sets (A, B, and C) of QC samples were prepared by spiking saliva samples from six independent donors with low QC (LQC) and HQC standards. Next, the samples were absorbed using the Mitra™ VAMS device (20 µL) and left to dry at ambient conditions for 2 h. For calculations, the ratios of chromatographic peaks of analyte and SIL-IS were used after determining the matrix-free samples (set A; phosphate-buffered saline replaced the matrix), and pre- and post-extraction spiked samples (sets B and C, respectively; with matrix spiked with analyte and SIL-IS). Assessed values were evaluated using the following mathematical formulas: ME = C/A × 100%; PE = B/A × 100%; and AR = B/C × 100%.

Stability testing, a crucial component of method development and validation, was conducted through various experiments, including storage stability in the autosampler and stability of analytes in VAMS tips under different stress conditions. Analyte stability is deemed acceptable if the differences stay within the nominal range of ±15%. Additionally, the reproducibility of the injection was assessed during autosampler stability testing.

For Incurred Sample Re-analysis (ISR) testing, 10% of the samples in the study should be re-analyzed. The ISR is calculated as the ratio of the difference between the re-analysis and initial results to the mean of these results. At least 67% of the re-analyzed samples must meet the acceptance criteria, which is within ±20% of the initial concentration value.

Cross-validation was used to compare the new dried saliva method with the traditional wet saliva method. The statistical analysis included Passing–Bablok regression, Bland–Altman bias estimation, and correlation analysis. For regression, the acceptance criterion was a slope within ±10% of the expected value (1.0), with both the intercept and slope falling within their respective confidence intervals around 0 and 1. Using the Bland–Altman method, the mean bias had to be less than ±20% (analytical limit of agreement, LoA) or less than ±15% (clinical limit of agreement, CoA) for at least 67% of the paired samples [21,22,23,24,25].

### 4.7. Evaluation of Saliva Sampling Correctness (Mitra™ VAMS Device)

Since saliva is a transparent biological matrix, ensuring sampling accuracy with the Mita™ microsampling device (Neoteryx, Torrance, CA, USA) is more challenging compared to capillary blood microsampling. To address this, the absorbed matrix was weighed following the procedure described by Mercolini et al. and compared with the weight of traditional wet saliva [5]. The scale used has a readability of 0.001 mg. For the tests, six separate saliva samples collected with Salivette^®^ systems were analyzed. First, 20 μL of wet saliva was pipetted into a 1.5 mL Eppendorf-type tube and weighed. The actual saliva weight was determined by subtracting the weight of the empty tube (measured before saliva transfer). 

Additionally, the density of the pipetted saliva was calculated as follows: saliva density (mg/mL) = mean weight of pipetted saliva (g)/0.02 mL (volume of pipetted matrix). Similarly, each VAMS tip (n = 10) was weighed before being absorbed in saliva. The sample was then gently absorbed to fill the polymer (+2 s) visually, and the loaded sampler was weighed again. The mass of the collected saliva was calculated by subtraction, and the density was determined similarly to that of wet saliva but using 24.17 μL as the average wicking volume for water-based solutions, based on the certificate of conformance provided by the manufacturer for each lot of Mita™ VAMS devices (LOT 37103-1).

### 4.8. Statistical Analysis

Validation calculations were performed using Excel version 16.91 (Microsoft, Redmond, WA, USA). Chromatographic peak counting and analysis were conducted with MultiQuant version 3.0.3 (AB Sciex, Framingham, MA, USA). Results are presented as mean ± SD and, where applicable, as a percentage coefficient of variation (CV%). Cross-validation and clinical validation were carried out using MedCalc software version 23.1.6 (MedCalc Software Ltd., Ostend, Belgium) for regression, bias, and correlation analyses. Normality was assessed with the Shapiro–Wilk test. Correlations of the data were examined using Pearson or Spearman correlation analysis, depending on whether the data were normally or non-normally distributed. Statistical significance was set at *p* < 0.05, with *p* < 0.0001 indicating significance in correlation analysis.

## 5. Conclusions

The novel method for determining MPA and MPAG in saliva collected using the VAMS device (20 μL; Mitra™) has been successfully optimized, validated, and clinically applied. Furthermore, the equivalence between dried saliva and conventional wet saliva has been demonstrated in cross-validation analysis. Saliva has been a subject of interest as an alternative to blood collection by venipuncture. The primary advantage of saliva sampling is its non-invasive nature, which allows for more frequent sampling and enables convenient self-sampling. Saliva sampling offers a significantly lower cost per sample, and salivary drug levels measurements are much easier and faster than quantifying the free drug concentration in plasma. Although validation parameters were satisfactory, the results confirm that saliva is not an entirely reliable surrogate matrix for routine TDM of MPA and its glucuronide, regardless of the sampling method. On the other hand, the demonstrated interchangeability between wet and dried saliva opens the possibility of further standardization of population studies based on saliva as an alternative matrix in therapeutic drug monitoring. Although saliva theoretically reflects the unbound drug fraction, the weak–modest association between salivary and plasma MPA/MPAG observed in this study—likely influenced by factors such as high MPA protein binding, salivary pH variation, flow rate, and oral microbiota—indicates that salivary concentrations do not consistently parallel plasma exposure in a clinically actionable manner. Capillary blood remains a practical and minimally invasive alternative, particularly in pediatric patients. Future research should further investigate the role of saliva in MPA monitoring, considering diverse clinical settings, patient populations, and physiological factors that influence the distribution of MPA in saliva.

## Figures and Tables

**Figure 1 pharmaceuticals-18-01744-f001:**
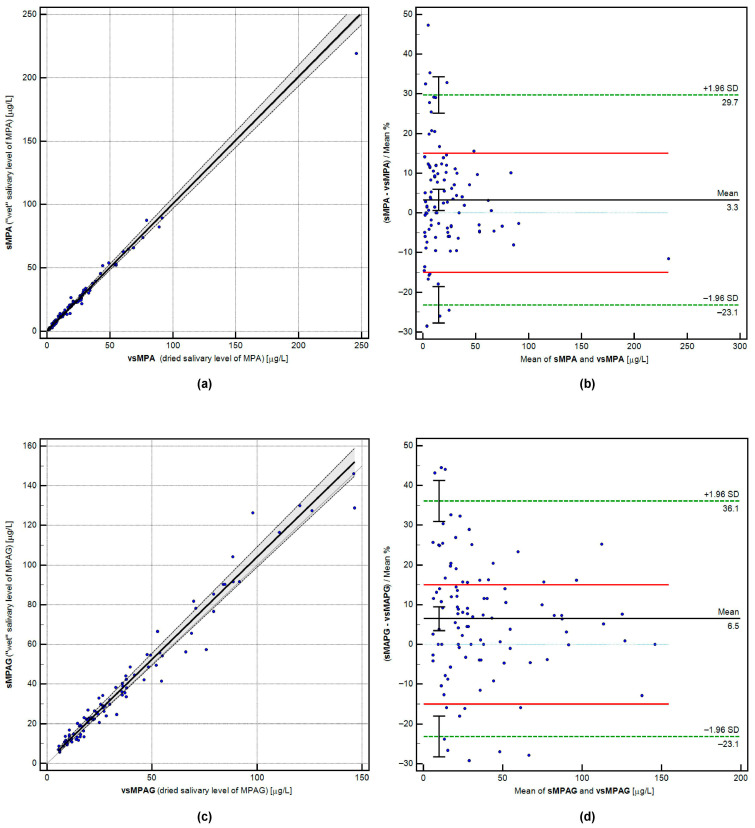
Visualization of cross-validation calculations for paired results (n = 100) of MPA and MPAG concentrations in wet and dried saliva samples. Panels (**a**,**c**) show Passing-Bablok regression for MPA and MPAG relationships (full description of comparison on plot axis). Panels (**b**,**d**) display Bland-Altman plots with the mean percentage bias marked, the standard deviation range (green dashed lines), and the analytical limit of agreement (LoA, red solid lines).

**Table 1 pharmaceuticals-18-01744-t001:** Summarized results of accuracy and precision evaluation for vsMPA and vsMPAG in dried saliva samples collected using the Mitra™ microsampling device (n = 6).

Level	Accuracy ^a^				Precision ^a^			
	Within-Run		Between-Run		Within-Run		Between-Run	
	*vsMPA*	*vsMPAG*	*vsMPA*	*vsMPAG*	*vsMPA*	*vsMPAG*	*vsMPA*	*vsMPAG*
LLOQ 1.00 μg/L	1.015 ± 0.097 104.13%	1.032 ± 0.113 109.64%	1.029 ± 0.077 103.48%	1.140 ± 0.088 105.55%	9.62%	10.94%	7.49%	7.81%
LQC 1.50 μg/L	1.545 ± 0.127 103.98%	1.411 ± 0.110 94.01%	1.505 ± 0.098 101.84%	1.515 ± 0.779 103.67%	8.24%	7.82%	6.53%	5.14%
MQC_1_ 5.00 μg/L	5.283 ± 0.417 106.65%	5.091 ± 0.258 103.88%	5.243 ± 0.162 104.24%	4.802 ± 0.190 97.78%	7.90%	5.06%	3.09%	3.95%
MQC_2_ 50.00 μg/L	50.667 ± 3.327 103.71%	52.664 ± 2.905 105.25%	50.283 ± 2.582 101.89%	50.890 ± 1.265 102.14%	6.57%	5.52%	5.10%	2.59%
HQC 500.00 μg/L	496.167 ± 12.624 98.83%	498.677 ± 13.556 97.28%	506.765 ± 4.844 100.99%	513.312 ± 9.873 101.14%	2.54%	2.72%	0.96%	1.92%

^a^ Data expressed as mean concentration with SD, mean percent value for accuracy and CV% for precision. LLOQ—lower limit of quantification; LQC—low quality control; MQC—medium quality control (1st and 2nd level); HQC—high quality control; vsMPA—mycophenolic acid level in dried saliva sample collected using volumetric absorptive microsampling; vsMPAG—mycophenolic acid glucuronide level in dried saliva sample collected using volumetric absorptive microsampling.

**Table 2 pharmaceuticals-18-01744-t002:** Summary of clinical and demographic data for the patients enrolled in the current study.

Demographic Parameter	Mean (SD; Range)
age [years]	13.64 (2.89; 10.50–17.08)
time after KTx [years]	3.15 ± 2.46 (1.50–15.25)
sex [boys/girls]	12/8
height [cm]	156.66 (13.18; 108.50–184.00)
weight [kg]	51.59 (12.34; 29.00–84.00)
MPA formulation (CellCept^®^)	20
MMF daily dose [mg]	500 (120–2000) ^a^
TAC formulations (Prograf^®^/Advagraf^®^)	13/7
TAC daily dose (Prograf^®^)	3.50 (1.00–9.50) ^a^
total protein in serum [g/L]	71.30 (5.67; 63.00–83.00)
albumin in serum [g/L]	45.30 (3.19; 38.00–51.00)
C-reactive protein (CRP) [mg/dL]	0.32 (0.97; 0.10–6.50)
serum urea [mg/dL]	39.40 (15.60; 19.00–89.40)
serum uric acid [mg/dL]	5.78 (1.12; 2.90–8.40)
serum creatinine [mg/dL]	0.95 (0.29; 0.47–1.76)
hematocrit [%]	37.60 (4.25; 28.40–48.20)

^a^ Expressed as median with range. KTx—kidney (renal) transplantation; MPA—mycophenolic acid; MMF—mycophenolate mofetil; TAC—tacrolimus.

**Table 3 pharmaceuticals-18-01744-t003:** Statistical results of cross-validation between sMPA and sMPAG levels in ‘wet’ saliva and vsMPA and vsMPAG in dried saliva collected using the Mitra™ microsampling device.

Statistical Parameter	sMPA vs. vsMPA	sMPAG vs. vsMPAG
Number of paired results	100	100
Mean bias (95% CI)	3.30% (0.62–5.97)	6.49% (3.49–9.49)
lower LoA [%]	−23.12%	−28.13%
upper LoA [%]	29.72%	36.11%
% of paired samples within ±20% bias (LoA)	97%	78%
% of paired samples within ±15% bias (CoA)	80%	66%
Passing-Bablok regression formula	sMPA = 1.00(vsMPA) + 0.17	sMPAG = 1.03(vsMPAG) + 0.95
regression: intercept (A)	0.1656 (−0.1122–0.7092)	0.9495 (−0.0159–1.9748)
regression: slope (B)	1.0047 (0.9686–1.0473)	1.0315 (0.9897–1.0721)
RSD [%]	2.59	4.09
SRCC	0.990 (*p* < 0.0001)	0.981 (*p* < 0.0001)
95% CI SRCC	0.9860–0.9940	0.9720–0.9870

CI—confidence interval; CoA—clinical limit of agreement (bias < 15%); LoA—analytical limit of agreement (bias < 20%); RSD—Relative Standard Deviation; SRCC—Spearman’s rank correlation coefficient; sMPA—salivary mycophenolic acid level; sMPAG—salivary mycophenolic acid glucuronide level; vsMPA—‘dried’ salivary mycophenolic acid level in sample collected with Mitra™; vsMPAG—‘dried’ salivary mycophenolic acid glucuronide level in sample collected with Mitra™.

**Table 4 pharmaceuticals-18-01744-t004:** Summarized results of the non-compartmental pharmacokinetic analysis of MPA and MPAG levels in the tested matrices.

	C_max_	T_max_ [h]	AUC_(0–4)_	AUC_(0–4)_ Normalized to Drug Dose
fMPA [mg/L]	0.16 (0.023–0.57)	0.90 (0.50–2.00)	0.276 (0.032–0.674) ^b^	0.596 (0.042–1.577) ^d^
tMPA [mg/L]	14.37 (6.14–36.65)	0.78 (0.50–1.00)	23.80 (11.46–70.05) ^b^	83.67 (22.44–447.22) ^d^
sMPA [µg/L]	83.54 (13.23–82.35)	0.85 (0.50–2.00)	106.68 (24.22–696.59) ^a^	300.18 (32.29–2716.80) ^c^
vsMPA [µg/L]	84.13 (13.28–87.13)	0.85 (0.50–2.00)	69.10 (23.68–157.73) ^a^	305.78 (33.88–2937.03) ^c^
vMPA [mg/L]	12.86 (4.87–34.02)	0.78 (0.50–1.00)	21.37 (10.82–63.34) ^b^	68.95 (20.71–444.79) ^d^
fMPAG [mg/L]	8.98 (3.85–19.81)	1.55 (0.50–2.00)	25.18 (11.62–51.39) ^b^	66.47 (19.97–195.68) ^d^
tMPAG [mg/L]	49.54 (19.24–109.28)	1.33 (0.50–2.00)	144.04 (65.62–288.73) ^b^	384.94 (106.81–1050.08) ^d^
sMPAG [µg/L]	73.74 (22.00–146.00)	1.08 (0.50–2.00)	157.71 (59.17–334.54) ^a^	352.10 (119.88–857.42) ^c^
vsMPAG [µg/L]	70.30 (19.04–146.54)	1.03 (0.50–2.00)	145.38 (55.75–319.11) ^a^	335.45 (115.15–902.00) ^c^
vMPAG [mg/L]	47.73 (19.80–110.76)	1.38 (0.50–2.00)	132.89 (67.52–234.08) ^b^	361.35 (100.49–1064.33) ^d^

^a^ µg × h/mL; ^b^ mg × h/mL; ^c^ µg × h/mL × g; ^d^ mg × h/mL × g; fMPA—free mycophenolic acid level; fMPAG—free mycophenolic acid glucuronide level; tMPA—total mycophenolic acid level; tMPAG—total mycophenolic acid glucuronide level; sMPA—wet salivary mycophenolic acid level; sMPAG—wet salivary mycophenolic acid glucuronide level; vsMPA—‘dried’ salivary mycophenolic acid level; vsMPAG—‘dried’ salivary mycophenolic acid glucuronide level; vMPA—capillary blood (VAMS) mycophenolic acid level; vMPAG—capillary blood (VAMS) mycophenolic acid glucuronide level; C_max_—maximum concentration; T_max_—time to maximum concentration; AUC_(0–4)_—area under the concentration to time curve.

**Table 5 pharmaceuticals-18-01744-t005:** Assigned nominal concentrations of calibrators (CCs) and quality controls (QCs) used for MPA and MPAG determination in dried saliva VAMS samples.

	CC1 (LLOQ)	QC1 (LQC)	CC2	QC2 (MQC_1_)	CC3	CC4	QC3 (MQC_2_)	CC5	CC6	CC7	QC4 (HQC)	CC8
MPA [µg/L]	1.00	1.50	2.00	5.00	10.00	25.00	50.00	75.00	150.00	350.00	500.00	700.00
MPAG [µg/L]	1.00	1.50	2.00	5.00	10.00	25.00	50.00	75.00	150.00	350.00	500.00	700.00

CC—calibration control; (L/M/H)QC—(low/medium/high) quality control; LLOQ—lower limit of quantification; MPA—mycophenolic acid; MPAG—mycophenolic acid glucuronide.

## Data Availability

The data supporting the findings of this study are available from the corresponding author upon reasonable request.

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
