# Peer review of "Volumetric Absorptive Microsampling of Saliva for Pharmacokinetic Evaluation of Mycophenolic Acid and Its Glucuronide Metabolite in Pediatric Renal Transplant Recipients: Bioanalytical Method Validation and Clinical Feasibility Evaluation"

_pharmaceuticals, 2025, doi:10.3390/ph18111744_

Round 1

Reviewer 1 Report

Comments and Suggestions for Authors

In the present manuscript, the authors demonstrated a novel volumetric absorptive microsampling method for estimating mycophenolic acid and its metabolite in plasma and saliva. The authors mentioned that this be useful on TDM considering salivary estimations. In general the manuscript is written well and work is extensive. Below are few comments for author’s attention:

Introduction

  • Lines 53: The authors may want to clarify what is stimulated and non-stimulated saliva.
  • Line 77: Typically, the adjustments of doses and optimization is based on plasma concentrations assuming that the plasma concentrations are surrogate for tissue concentrations. The authors my want to explain how saliva concentration can be used for this purpose.
  • Lines 100: Please clarify what the authors meant by correlation between plasma concentrations and saliva concentrations for MPA? Do they mean increase of plasma concentrations led to saliva concentrations and what factor difference may be anticipated? Please also clarify if MPA determination is suitable in saliva and what factors may play a role in such estimations.
  • The authors may want to provide additional details or description about MitraTM device for better understanding.
  • The authors may want to elaborate on the existing literature for plasma and saliva estimations for MPA and metabolite to demonstrate novelty of this work.

Materials and methods

  • Section 4.3.4: Please clarify the meaning of capillary blood.
  • Section 4.5: few details such as pediatric age group, administration protocols (how many days, dosing was performed to reach to steady state) were not provided. The time points for saliva collection are not provided. Also details for stimulated saliva collection are not provided.
  • Section 4.6: the calibration curve ranges for both plasma and saliva may need to be clarified better. What is comparison of calibration curves in both cases, since the authors are determining concentration of drug and metabolite in various biological matrices, I suggest that the authors can include a tabular comparison of all 2 analytes for all biological matrices along with LC-MS/MS transitions and conditions for better clarity.

Results

  • Section 2.1: The rationale for time points selection in saliva for drying needs to be clarified.
  • I also suggest that similar to Figure 1, the authors may want to show correlation between saliva and plasma levels of drug and metabolite for better explanation and interpretation.
  • From the provided plasma concentration time profiles, it seems the elimination is not reached. The authors may want to explain why the sampling is limited to 4 h and reasons for not extending the same.

Author Response

We would like to thank the Reviewer for the careful evaluation of our manuscript and valuable comments. All suggested revisions have been implemented and significantly improving the clarity and quality of the paper. Below we respond point-by-point:

[1] Reviewer Comment 1: Lines 53: The authors may want to clarify what is stimulated and non-stimulated saliva. 

[1] Author's Response: Thank you. We expanded this paragraph to define both types of saliva and the collection procedures clearly. This content is now included in the Introduction (Page 2).

[2] Reviewer Comment 2: Line 77: Typically, the adjustments of doses and optimization is based on plasma concentrations assuming that the plasma concentrations are surrogate for tissue concentrations. The authors my want to explain how saliva concentration can be used for this purpose.

[2] Author's Response: We clarified that saliva theoretically reflects the unbound drug fraction and may be used to predict plasma exposure through matrix-bridging studies. Additional explanation and references concerning the pharmacological relevance of the free drug fraction were added (Page 3).

[3] Reviewer Comment 3: Lines 100: Please clarify what the authors meant by correlation between plasma concentrations and saliva concentrations for MPA? Do they mean increase of plasma concentrations led to saliva concentrations and what factor difference may be anticipated? Please also clarify if MPA determination is suitable in saliva and what factors may play a role in such estimations.

[3] Author's Response: We revised the text to explain expected associations and influencing factors (pH, protein binding, saliva flow rate, matrix physiology). We also highlighted the controversies in previous studies (Pages 3–4).

[4] Reviewer Comment 4: The authors may want to provide additional details or description about MitraTM device for better understanding.

[4] Author's Response: The device description was expanded to include polymer tip technology, volumetric accuracy, and advantages for sample stability (Page 2).

[5] Reviewer Comment 5: The authors may want to elaborate on the existing literature for plasma and saliva estimations for MPA and metabolite to demonstrate novelty of this work.

[5] Author's Response: We have added a broader literature overview on saliva-based MPA monitoring in adult and pediatric cohorts, clarifying that this is the first study to implement VAMS for MPA/MPAG saliva sampling in a clinical setting (Pages 4–5).

[6] Reviewer Comment 6: Section 4.3.4: Please clarify the meaning of capillary blood.

[6] Author's Response: We clarified the term as blood from skin capillaries collected via a fingerprick using VAMS (Page 18).

[7] Reviewer Comment 7: Section 4.5: few details such as pediatric age group, administration protocols (how many days, dosing was performed to reach to steady state) were not provided. The time points for saliva collection are not provided. Also details for stimulated saliva collection are not provided.

[7] Author's Response: These details are now clearly stated in the M&M section and Results as well: pediatric age 10.5–17 years; steady-state MMF treatment since transplantation; saliva collected at 0, 0.5, 1, 2, 4 h post-dose; only unstimulated saliva was used (stimulated saliva was excluded due to tolerability) (Page 20).

[8] Reviewer Comment 8: Section 4.6: the calibration curve ranges for both plasma and saliva may need to be clarified better. What is comparison of calibration curves in both cases, since the authors are determining concentration of drug and metabolite in various biological matrices, I suggest that the authors can include a tabular comparison of all 2 analytes for all biological matrices along with LC-MS/MS transitions and conditions for better clarity.

[8] Author's Response: Since the primary objective of the study was to validate the novel dried saliva-based LC-MS/MS method, we decided not to include a detailed tabular comparison of matrices and methods. In the case of other matrices, we relied on previously published studies. Additionally, other reviewers suggested limiting the analytical and validation background of the study. 

[9] Reviewer Comment 9: Section 2.1: The rationale for time points selection in saliva for drying needs to be clarified.

[9] Author's Response: To clarify, the drying time for collected saliva has been optimized through experiments and is detailed in Section 2.1. Suppose the Reviewer was referring to the choice of time points for saliva collection after drying. In that case, this has now been adequately described in the Methods & Materials sections and the Discussion section. 

[10] Reviewer Comment 10: I also suggest that similar to Figure 1, the authors may want to show correlation between saliva and plasma levels of drug and metabolite for better explanation and interpretation.

[10] Author's Response: The correlation between each matrix is presented in the manuscript as a correlation study in the results. We decided not to include regression studies and bias estimation between matrices other than wet and dried saliva due to limited clarity of data presentation in Bland-Altman and Passing-Bablok plots (high disproportions). On the other hand, the correlation study has been discussed in greater detail in the Results and Discussion sections. 

[11] Reviewer Comment 11: From the provided plasma concentration time profiles, it seems the elimination is not reached. The authors may want to explain why the sampling is limited to 4 h and the reasons for not extending the same.

[11] We included justification for pediatric feasibility and ethical considerations, aligned with the standard clinical visit schedule. The focus on the early exposure phase for matrix bridging was emphasised, consistent with prior pediatric studies. A detailed justification has been added to the Discussion (Page 27).

Reviewer 2 Report

Comments and Suggestions for Authors

General Comments

Overall, the manuscript presents a novel and clinically relevant concept - the use of Mitra™ VAMS devices for saliva (oral fluid) sampling to determine MPA and MPAG concentrations for TDM in pediatric renal transplant patients. The study is well-designed, the analytical validation appears solid, and the clinical application is meaningful. The work adds important knowledge to the field of microsampling and non-invasive TDM.

However, the organization and writing style of the manuscript need significant revision before it can be considered for publication. At present, the paper reads more like a laboratory report than a structured scientific article. The results are overloaded with raw numerical data, and the discussion lacks focus in several areas. To improve readability and impact, the manuscript needs restructuring, better flow, and reduction of redundancy.

I would suggest better separation and ordering of the sections in methods and results. Some technical details (such as validation parameters) could be moved to the Supplementary Material.

The Results section currently lists too many numerical values (for example, correlation data in lines 229-240) without explaining what they mean biologically or clinically. Summarize the key findings in text and place the detailed numbers in a table or supplement.

The Discussion section repeats some information already presented in the Results. Please reduce overlap and focus more on interpretation and clinical implications.

The paper contains excessive background and validation detail that makes it difficult for readers to follow the main message. Try to simplify wherever possible. Remove/Move less critical or repetitive information to supplementary.

Specific comments:

  1. Include more context explaining why MPA is an important or challenging drug for saliva-based TDM in the introduction section. Mention that MPA has a narrow therapeutic index, high interindividual variability, and that TDM is critical for transplant patients to maintain efficacy and prevent toxicity.
  2. While authors have highlighted that this is “the first study to use VAMS for MPA and MPAG in saliva,” the existing research gap is not clearly described. Please specify what was missing in earlier studies and how this study is filling those knowledge gaps.
  3. Use either saliva or oral fluid (OF) consistently throughout the manuscript.
  4. Check that all abbreviations (VAMS, qDBS, MPA, MPAG, tMPA, fMPA, etc.) are clearly defined at first appearance.

Author Response

We would like to thank the Reviewer for the careful evaluation of our manuscript and valuable comments. All suggested revisions have been implemented and significantly improving the clarity and quality of the paper. Below we respond point-by-point:

[1] Reviever Comment 1: Overall, the manuscript presents a novel and clinically relevant concept - the use of Mitra™ VAMS devices for saliva (oral fluid) sampling to determine MPA and MPAG concentrations for TDM in pediatric renal transplant patients. The study is well-designed, the analytical validation appears solid, and the clinical application is meaningful. The work adds important knowledge to the field of microsampling and non-invasive TDM. However, the organization and writing style of the manuscript need significant revision before it can be considered for publication. At present, the paper reads more like a laboratory report than a structured scientific article. The results are overloaded with raw numerical data, and the discussion lacks focus in several areas. To improve readability and impact, the manuscript needs restructuring, better flow, and reduction of redundancy. I would suggest better separation and ordering of the sections in methods and results. Some technical details (such as validation parameters) could be moved to the Supplementary Material. The Results section currently lists too many numerical values (for example, correlation data in lines 229-240) without explaining what they mean biologically or clinically. Summarize the key findings in text and place the detailed numbers in a table or supplement. The Discussion section repeats some information already presented in the Results. Please reduce overlap and focus more on interpretation and clinical implications. The paper contains excessive background and validation detail that makes it difficult for readers to follow the main message. Try to simplify wherever possible. Remove/Move less critical or repetitive information to supplementary.

[1] Authors' Response: The manuscript has been reorganised for better clarity. Major updates involve summarising numerical results in tables and supplements, streamlining the narrative, eliminating redundant descriptive sections, and moving some validation details to the Supplementary Materials.

[2] Reviever Comment 2: Include more context explaining why MPA is an important or challenging drug for saliva-based TDM in the introduction section. Mention that MPA has a narrow therapeutic index, high interindividual variability, and that TDM is critical for transplant patients to maintain efficacy and prevent toxicity.

[2] Author's Response: We expanded the Introduction to emphasize MPA’s narrow therapeutic index, high PK variability, and importance of TDM in pediatric transplantation (Page 4).

[3] Reviever Comment 3: While authors have highlighted that this is “the first study to use VAMS for MPA and MPAG in saliva,” the existing research gap is not clearly described. Please specify what was missing in earlier studies and how this study is filling those knowledge gaps.

[3] We have added a broader literature overview on saliva-based MPA monitoring in adult and pediatric cohorts, clarifying that this is the first study to implement VAMS for MPA/MPAG saliva sampling in a clinical setting (Pages 4–5).

[4] Reviever Comment 4: Use either saliva or oral fluid (OF) consistently throughout the manuscript.

[4] Author's Response: We standardized the use of “saliva” (instead of mixed “oral fluid/saliva”). 

[5] Reviever Comment 5: Check that all abbreviations (VAMS, qDBS, MPA, MPAG, tMPA, fMPA, etc.) are clearly defined at first appearance.

[5] Author's Response: All abbreviations are now defined at first appearance.

Reviewer 3 Report

Comments and Suggestions for Authors

This study explored the use of saliva as a non-invasive alternative to plasma for monitoring Mycophenolic acid (MPA) and its primary metabolite, MPAG, in pediatric renal transplant patients. It found that while dried and wet saliva exhibited high internal correlations, they showed poor correlation with both unbound and total plasma concentrations of MPA and MPAG. The study concluded that saliva, despite being minimally invasive, is not a reliable substitute for plasma in routine therapeutic drug monitoring (TDM) of MPA. Capillary blood collected via Volumetric Absorptive Microsampling (VAMS) is a more promising alternative.

"Despite extensive efforts to investigate saliva as a non-invasive alternative for monitoring MPA and MPAG levels, the results consistently showed weak to modest correlations with plasma concentrations. As a result, saliva sampling, although minimally invasive and patient-friendly, has not been accepted as a reliable method for routine therapeutic drug monitoring in pediatric renal transplant patients. These findings highlight the ongoing challenges in finding an accurate and practical alternative to plasma for drug monitoring, especially in populations requiring long-term, precise management."

Author Response

Reviewer Comments: This study explored the use of saliva as a non-invasive alternative to plasma for monitoring Mycophenolic acid (MPA) and its primary metabolite, MPAG, in pediatric renal transplant patients. It found that while dried and wet saliva exhibited high internal correlations, they showed poor correlation with both unbound and total plasma concentrations of MPA and MPAG. The study concluded that saliva, despite being minimally invasive, is not a reliable substitute for plasma in routine therapeutic drug monitoring (TDM) of MPA. Capillary blood collected via Volumetric Absorptive Microsampling (VAMS) is a more promising alternative.

Despite extensive efforts to investigate saliva as a non-invasive alternative for monitoring MPA and MPAG levels, the results consistently showed weak to modest correlations with plasma concentrations. As a result, saliva sampling, although minimally invasive and patient-friendly, has not been accepted as a reliable method for routine therapeutic drug monitoring in pediatric renal transplant patients. These findings highlight the ongoing challenges in finding an accurate and practical alternative to plasma for drug monitoring, especially in populations requiring long-term, precise management.

Authors' Response: We thank the Reviewer (3) for this precise and helpful comment. We agree that, despite significant efforts to explore saliva as a non-invasive medium for monitoring mycophenolic acid (MPA) and its glucuronide metabolite (MPAG), our results show only weak to moderate correlations between salivary and plasma levels, whether total or unbound. Consistent with existing research, these results suggest that, although saliva is convenient and minimally invasive, it currently lacks the analytical accuracy and clinical reliability needed for routine TDM of MPA in pediatric kidney transplant patients.

Our goal was to thoroughly examine this matrix in a controlled clinical setting by validating a reliable LC–MS/MS method, testing both wet and dried saliva samples collected via VAMS, and comparing these results with those obtained from traditional plasma and capillary microsampling. While the internal consistency across saliva samples was excellent, the weak correlation with plasma concentrations and pharmacokinetics highlights the current limitations of using saliva in this context.

Furthermore, our data underscore the clinical value of capillary blood VAMS as a more precise and practical alternative to venous sampling in this patient group. Its high acceptability among patients and strong agreement with plasma results support its potential for inclusion in future pediatric TDM protocols.

We thank the Reviewer for recognizing these challenges, and we believe our findings help clarify the limitations as well as the opportunities of non-invasive sampling for personalized immunosuppression management.

Round 2

Reviewer 1 Report

Comments and Suggestions for Authors

I would like to thank the authors for making necessary corrections in the manuscript and clarifying the comments.

Reviewer 2 Report

Comments and Suggestions for Authors

I am satisfied with Authors responses and current version of manuscript is acceptable for publication in Pharmaceuticals.